# Interactive System for Similarity-Based Inspection and Assessment of the Well-Being of mHealth Users

**DOI:** 10.3390/e23121695

**Published:** 2021-12-17

**Authors:** Subash Prakash, Vishnu Unnikrishnan, Rüdiger Pryss, Robin Kraft, Johannes Schobel, Ronny Hannemann, Berthold Langguth, Winfried Schlee, Myra Spiliopoulou

**Affiliations:** 1Knowledge Management and Discovery Lab, Otto-von-Guericke University, 39106 Magdeburg, Germany; 2Institute of Clinical Epidemiology and Biometry, University of Würzburg, 97078 Würzburg, Germany; ruediger.pryss@uni-wuerzburg.de; 3Institute of Databases and Information Systems, Ulm University, 89081 Ulm, Germany; robin.kraft@uni-ulm.de; 4Institute DigiHealth, Neu-Ulm University of Applied Sciences, 89231 Neu-Ulm, Germany; johannes.schobel@hnu.de; 5WSAudiology, Sivantos GmbH, 91058 Erlangen, Germany; ronny.hannemann@sivantos.com; 6Department of Psychiatry and Psychotherapy, University of Regensburg, 93053 Regensburg, Germany; berthold.langguth@medbo.de (B.L.); winfried.schlee@tinnitusresearch.org (W.S.)

**Keywords:** medical analytics, condition prediction, ecological momentary assessment, visual analytics, time series

## Abstract

Recent digitization technologies empower mHealth users to conveniently record their Ecological Momentary Assessments (EMA) through web applications, smartphones, and wearable devices. These recordings can help clinicians understand how the users’ condition changes, but appropriate learning and visualization mechanisms are required for this purpose. We propose a web-based visual analytics tool, which processes clinical data as well as EMAs that were recorded through a mHealth application. The goals we pursue are (1) to predict the condition of the user in the near and the far future, while also identifying the clinical data that mostly contribute to EMA predictions, (2) to identify users with outlier EMA, and (3) to show to what extent the EMAs of a user are in line with or diverge from those users similar to him/her. We report our findings based on a pilot study on patient empowerment, involving tinnitus patients who recorded EMAs with the mHealth app TinnitusTips. To validate our method, we also derived synthetic data from the same pilot study. Based on this setting, results for different use cases are reported.

## 1. Introduction

Ecological Momentary Assessments (EMAs) are utilized to capture the immediate behavioral experience for a medical phenomenon. The behavioral experience can range from the present moment to a few minutes earlier, or can be a recollection of events that occurred at earlier, longer time frames. Presently, EMAs are mainly recorded with help of mobile technology, namely, by digital devices that notify the users multiple times for a period of days or weeks, so that they record current or recent medical states, behaviors, or environmental conditions [1]. By using this approach, the period of recall can be reduced to hours or minutes. Hence, EMAs allow observing the natural set of behaviors and moods of the participants [2]. EMAs are used for behavioral monitoring and tracking the progression of bipolar disorders [3,4], for studying tinnitus distress [5], and for monitoring mood in major depressive disorders [6]. Importantly, EMAs reduce the recall bias [7,8,9], and can thus be exploited by physicians for reliable patient monitoring and decision support. Comprehensible Artificial Intelligence (cAI) [10] is a transition framework that encompasses multiple disciplines such as AI, Human-Computer Interaction (HCI), and End User explanations, along with the combinations of techniques and approaches such as visual analytics, interactive ML, and dialog systems. Visual analytics act as an intersection between AI and end-user explanations to provide rich visualizations, which are helpful for humans to understand, interpret and further improve the trust of a developed system. A Clinical Decision Support System (CDSS) involves the provision of effective assistance to clinicians during the process of patient treatment and diagnosis [11]. It was utilized for effective and interactive communication between a physician and patients through alerts that are provided during self-monitoring [12]. The ability for the patients to make clinical decisions are also utilized in CDSS [13]. Recently, Intellicare [14] platforms attempt to provide remote therapies for anyone at any point in their mental health journey through smartphone-based applications to reduce stress, depression, and anxiety. By including the parts of cAI transition framework, the existing systems can be enhanced.

Note that considerations of mHealth data in the context of well-being are not fundamentally new. Studies and concepts can be found that deal with the combined perspective of well-being and mHealth. In [15], for example, a technical concept is discussed, in which mHealth monitoring is related to well-being monitoring in general, concretely by a comparison with an online social network scenario. The results shown by [16], in turn, reveal that emotional bonding with mHealth apps can be related to the well-being of the users. Many other recent and related works can be found in this context [17,18,19]. However, the decision to investigate the similarity among the users to assess their well-being is not considered so far, to the best of the authors’ knowledge. In addition, to accomplish the similarity inspection based on advanced visualization techniques is not being pursued in the same way by other works.

In this work, we contribute to CDSS through a medical analytics interactive tool that facilitates the inspection of a user’s EMA and clinical data. It juxtaposes timestamped EMA recordings of users with related clinical data, and predicts future user recordings with respect to symptoms of interest. In particular, we investigate the following research questions:**RQ1-** How to predict a user’s EMA in the near and far future, on the basis of similarities to other users?**RQ2-** How to identify and show outlierness in users’ EMAs?**RQ3-** How to show similarities and differences in the EMAs of users who are similar in their clinical data?

The core idea of our approach is the exploitation of similarity among users in their static clinical data as well as in their dynamic, timestamped EMAs. We build upon similarity for prediction in time series (see RQ1), upon identification of users with outlier behavior (see RQ2), and upon visual and quantitative juxtaposition among users (see RQ3). For the validation of our work, we use the data of a pilot study on the role of mHealth tools for patient empowerment. The study involved 72 tinnitus patients, who recorded EMAs with the mHealth tool TinnitusTipps over an 8-weeks period between 2018 and 2019.

Our contributions can be summarized as follows:We demonstrate the users neighborhood comparisons over data (i.e., both static, dynamic, and timestamped EMA) and utilize them for predicting user’s EMA recordings and show that users neighborhood are indeed useful in making the ahead predictions.We introduce a voting-based outlier detection methodology to identify users who behave differently in their interaction with the app and also introduce tailored interactive visualizations that can be inspected.We introduce a medical analytics tool with the introduction of tailored interactive visualizations to demonstrate the nearest neighboring user’s behaviors recorded through the app, and a visualization to show the ahead predictions for a study based on the identified nearest neighbors by constructing pathways.

The remainder of this paper is structured as follows. In Section 2 we provide the necessary information about the mobile health app and data of the pilot study on TinnitusTipps. In Section 3, Section 5 and Section 6 we detail the methods introduced for answering our RQ’s. In Section 7 we report on the results of our analysis. In Section 8 we elaborate on the obtained findings and discuss improvements and limitations.

## 2. Materials

The tinnitus study was approved by the Ethical Review Board of the University Clinic Regensburg. The ethical approval number is 17-544-101. All study participants provided informed consent.

### 2.1. The TinnitusTipps Mobile Health App

The TinnitusTipps app (https://tinnitustipps.lenoxug.de/, accessed on 15 December 2021) was developed by computer scientists, psychologists, and the Sivantos GmbH (which is a company specialized in hearing aids). TinnitusTipps is based on two considerations. Firstly, we wanted to utilize and exploit the experiences and benefits of the TrackYourTinnitus (TYT) mobile app. TYT combines Ecological Momentary Assessments and Mobile Crowdsensing (EMA-MCS) [20] to perform Digital Phenotyping [21] for users affected by tinnitus. Digital Phenotyping, in turn, is based on the idea to perform Ecological Momentary Assessments by the use of digital technology like smartphones. Secondly, we wanted to improve TYT and therefore added feedback features. In TYT, we only gathered EMA data from participating users, but no feedback was sent back to them. Compared to TYT, TinnitusTipps contains following new features: daily tinnitus tipps, data visualizations, and direct feedback from a healthcare provider.

### 2.2. Data of the CHRODIS+ Pilot Study on Tinnitus

From TinnitusTipps we collected following types of data:*Registration data* with the Tinnitus Sample Case History Questionnaire (TSCHQ, 35 items) [22] and the Tinnitus Hearing Questionnaire (THQ, 6 questions) (cf. Appendix B
Table A1) completed by all app users.*EMA data:* with an EMA questionnaire (8 items) that captures within-day fluctuations of tinnitus loudness, distress, of hearing ability, stress and further aspects of mood and health condition (cf. Appendix B Table A2). As it can be seen in Appendix B Table A2, this questionnaire consists of 6 short questions to be answered with a numerical value between 0 and 100%, where larger values indicate worse condition.Users are notified by the smartphone multiple times within a day, and randomly, to record the current EMA.

For our analyses, we considered only users who have recorded both types of data, i.e., we excluded 19 users for which we did not have the registration data or who did not record any EMA. The total number of EMA recordings of the 53 users was 7255.

## 3. Methods

Our workflow consists of methods for data conversion (1), neighborhood creation (2), and generation of question combinations as features (3), which are used for all three RQs. The backend and frontend components of our workflow are shown in Figure 1 and explained hereafter.

On Table 1 we summarize the notation we used throughout this paper.

### 3.1. Data Conversion

The collected user data were stored in the JavaScript Object Notation (JSON) format, from which they were converted into Comma Separated Values (CSV) format. The processed data encompass two types of user answers: the *static data* from the two registration questionnaires and the *time series data* from the EMA questionnaire. The numeric values of the EMA variables were rescaled to be between 0 and 1. We used the time series on loudness (s02) and tinnitus distress (s03) for prediction, and the time series of the items s02, s03, …, s08 for visualizations.

### 3.2. Time Series Alignment

The users start recording their observations over different time periods and after a few days, there are possibilities to stop. To deal with such gaps and variations in the user’s time series recordings, a time series alignment approach is taken into account. The approach involves the aggregating the time series at a daily-level using either mean, maximum or minimum. The first observation within a given month is initialized to 0, and is assigned to the prev_encoded_val variable. For the next observation within the given month, the previous_encoded_val is incremented and the updated value is reassigned to previous_encoded_val. For the first observations for the next month, the last day index value of previous_encoded_val is obtained and incremented accordingly until the last recording for a user. For the ahead predictions, *minimum*, *maximum* and *mean* aggregations are investigated.

### 3.3. Creation of Neighborhood

We created two types of neighborhoods—using static data and using time series data, as described hereafter.

#### 3.3.1. Neighborhoods on Static Data

Let $U=\left[u_{1},u_{2},\ldots ,u_{n}\right]$ represent a set of users, with n being the number of users and features of a user $u_{n}$ is denoted by $staticF=\left[s_{1},s_{2},\ldots ,s_{r}\right]$, with |staticF| is the total number of features in the static data. The response attributes in staticF contain the user’s static information (age, background, and comorbidities) and their past medical conditions (presence/absence of neck pain, etc.).

For each user, we compute the set of k-nearest neighbors subject to a threshold, namely, the *average of the distances over all pairs of users*, so that only closer neighbors contribute to the similarity.

The Heterogeneous Euclidean Overlap Metric (HEOM) is used to measure the similarity. The definitions hereafter come from [23,24], but we use a different notation.

Let *s*_*r*_(*x*), *s*_*r*_(*z*) be the feature values of $s_{r}\in staticF$, then the overlap metric distance for the users x,z∈U is defined as: (1)$$d_{OM}\left(x,z\right)=\sum_{i=1}^{n}f_{OM}\left(s_{r}\left(x\right),s_{r}\left(z\right)\right)$$
with, $f_{OM}\left(s_{r}\left(x\right),s_{r}\left(z\right)\right)=\left\{\begin{array}{cc} 0 & \;\mathrm{if}\;s_{r}\left(x\right)=s_{r}\left(z\right)\\ 1 & \;\mathrm{otherwise}\; \end{array}\right.$

When the attributes value of *s*_*r*_(*x*), *s*_*r*_(*z*) are continuous, then we define the range difference between the two attribute values as follows: (2)$$rn\_diff_{s_{r}}\left(x,z\right)=\frac{|s_{r}\left(x\right)-s_{r}\left(z\right)|}{range_{s_{r}}}$$
where, $range_{s_{r}}=max\left(s_{r}\right)-min\left(s_{r}\right)$ is used to normalize the attributes.

From the overlap metric and range difference, we derive *HEOM* as: (3)$$d_{HEOM}\left(x,z\right)=\sqrt{\sum_{i=1}^{n}f_{HEOM}{\left(s_{r}\left(x\right),s_{r}\left(z\right)\right)}^{2}}$$
with, $f_{HEOM}\left(s_{r}\left(x\right),s_{r}\left(z\right)\right)=\left\{\begin{array}{cc} f_{OM}\left(s_{r}\left(x\right),s_{r}\left(z\right)\right) & \;\mathrm{if}\;s_{r}\;\mathrm{is}\;\mathrm{nominal}\;\\ s_{r}\left(x\right)-s_{r}\left(z\right) & \;\mathrm{if}\;s_{r}\;\mathrm{is}\;\mathrm{continuous}\\ 1 & \;\mathrm{otherwise}\; \end{array}\right.$

#### 3.3.2. Neighborhoods on the Dynamic Data of the EMA Time Series—One per EMA Item

Next to the static information, the users also record an ordered sequence of observations that constitute the EMA recordings. Let $U=\left[u_{1},u_{2},\ldots ,u_{n}\right]$ represent the set of users with the EMA recordings. We denote the sequence of observations of an $u_{n}\in U$ as $u_{n}=\left[o_{1},o_{2},\ldots ,o_{i},\ldots ,o_{m}\right]$, where each observation contains a set of features along with a timestamp $o_{i}$ = {$t_{i}$, $F_{i}$}, where $t_{i}$ represents the timestamp of the user $u_{n}$, $F_{i}$ represent the feature space of EMA recordings which is denoted as $F_{i}=\left[f_{1},f_{2},\ldots ,f_{j}\right]$,for $j=\left[1,2,\ldots ,|F_{i}|\right]$; $|F_{i}|$ = 8, and *m* represents the total number of EMA recordings recorded by $u_{n}$ over the days denoting the length of the time series. The multiple observed values recorded by the user within a given day for the considered EMA variable are averaged for the neighborhood computations.

We compute the similarity between a user *x* and user *z* for each EMA item separately, after aligning their time series at the same (nominal) day 0:‘Day matching’ between *x* and *z*: number of days from day 0 onward and until the last day of the shortest between the two time series. For example, if *x* has EMA observations for 30 days and *z* for 60 days, then the matching is on the first 30 days, with day k=0,…,29 of *x* matched to the corresponding day of *z*.For each of the matched days for *x* and *z*, a euclidean distance is computed. A counter is maintained to capture the number of days when both the users have reported their observations, denoted by, counts and the distance obtained for each day is summed up and is denoted as total_dist.Finally, a fraction of the $\frac{total\_dist}{counts}$ is returned as the similarity between the user *x* and *z*.

### 3.4. Creation of Feature Combinations

The similarity between users and, accordingly, their neighborhoods over static data, can be computed using the whole set of registration items or a subset of them. To assess the effect of different aspects of a user’s recordings, we construct the following overlapping ‘subspaces’, i.e., combinations of features from the set of registration items:C1:user background and tinnitus complaints information (items tschq02-04); tinnitus historical information (items tschq05-18), including the initial onset of tinnitus, loudness, tinnitus awareness, and different past treatmentsC2:experienced effects of tinnitus (tschq19-25), and questions on hearing quality/loss (hq01, hq03)C3:further conditions, such as neck pain, dizziness etc. (tschq28-35), as well as the items on hearing quality/hearing loss (hq02, hq04)C4:all the TSCHQ and HQ items

Along with these combinations, the numerical value loudness (tschq12) is also included, so that there are no ties in the computation of user similarity.

By using these combinations, a similarity is built for the specified neighborhood size (k); i.e., for each of the user and the obtained similar users are utilized for the prediction of tinnitus distress by computing the k nearest neighbors over the registration data.

## 4. RQ1: How to Predict a User’s EMA in the Near and Far Future, on the Basis of Similarity to Other Users?

A prediction of a class label for a test user *u*, for an EMA recording $f_{j}\in F$ at a timepoint $t_{i}$ is performed through obtaining the nearest neighbor users of *u* and by utilizing the nearest neighbors EMA recordings until $t_{i}$. The work focuses on the prediction of a numeric class label only.

### 4.1. Ahead Prediction of the Target Variable

Let T be the set of test users, let *u* be a test user, and let $t_{i}$ be the timepoint at which we want to predict the values of l≥1 future EMA recordings for *u*. We first compute the set of nearest neighbors of *u* for a given *k*. We denote it as $NN_{k}\left(u\right)$. Then, we consider the following options:*Weighted average in a user’s neighborhood:* At timepoint $t_{i}$, the predicted value of an EMA recording is then the average of the values of this EMA recording over the users in $NN_{k}\left(u\right)$. This computation can be done for l≥1 timepoints at once.*Linear regression in a user’s neighborhood:* for a test user u∈T, a linear regressor is built on the time series observations of each of the nearest neighbor users $u_{j}\in NN_{k}\left(u\right)$ until the timepoint and averaging the slope ($m_{slope,u_{j}}$) and the intercept: ($m_{intercept,u_{j}}$) parameters of each of the nearest neighbor user to create a combined model as introduced in [25].

### 4.2. Evaluation for RQ1

We investigate how the value of *k*, the choice of subspace (from the registration data), the kind of neighborhood (static vs dynamic) and the aggregation function over the EMA within a day (min, mean, max) affect the quality of the prediction. For the evaluation, we consider Root Mean Squared Error (RMSE) over a set T test users. In particular:For a test user $u_{i}\in T$, the nearest neighbors $NN_{k}\left(u_{i}\right)$ are obtained (either for static or dynamic). Once after obtaining the nearest neighbors of $u_{i}$, their respective time series are aligned, and wherever multiple observations are present for a given day, they are aggregated using either the minimum(), mean(), or maximum() functions.Next, based upon ahead prediction timepoint (in a day), future EMA recordings (l), and the nearest neighbors time series information; ahead predictions are obtained based on the proposed methodologies as per Section 4.1.For a test user, the error for a feature $f_{i}\in F$ is measured using Root Mean Squared Error (RMSE) against the true value of that feature as follows:
$$RMSE\left(f_{i},\hat{f_{i}}\right)=\sqrt{\frac{1}{l}\sum_{i=1}^{n}{\left(f_{i}-\hat{f_{i}}\right)}^{2}}$$
where $\hat{f_{i}}$ denotes the predicted values obtained for a feature $f_{i}\in F$, and *l* denotes the number of ahead days to predict.

Let the above three steps of evaluation be denoted as eval. The same process is carried out for all the test users and compute the final quality value for a given neighborhood size *k* as
$$avgRMSE\left(T,k\right)=\frac{1}{\left|T\right|}\left(\sum_{i=1}^{\left|T\right|}eval\left(u_{i}\right)\right)$$
and the achieved RMSE for all the test users is averaged for the provided neighborhood size (*k*).

To assess the changes in ahead-prediction from earlier days to days in the distant future, a comparison between near predictions (earlier days) and far predictions (far-future days) are also done based on the concepts of [25]. The neighborhood size from 1 to 30 is chosen in this work.

## 5. RQ2: How to Identify and Show Outlierness in User’s EMA

An outlier instance is one whose properties deviate from most of the other instances. We highlight our workflow next, and detail the concept of visualizing the tendency of outlierness in Section 6.2.

### Methodology

Our approach improves the rank-based outlier detection approach [26] using the voting-based strategy proposed in [27]. Instead of working over a single similarity measure and manually defining the neighborhood, a voting-based strategy with the capabilities to work over the tailored similarity measure is employed. The algorithm runs the detection over a set of *k* neighborhoods, over each of the defined subspace combinations and votes the outlying data object at each of the neighborhood iterations based upon the outlier score as per Appendix D Equation (A1) to pick out the top-n outliers, which are highest voted over the entire set of the defined neighborhoods. According to our proposed methodology, if the z-score value is greater than or equal to 2.0, then it is treated as an outlier since the algorithm runs over multiple values of *k* and the scores are combined as overall average scores across the set of the neighborhoods. Additionally, the proposed voting strategy also utilizes the Local Outlier Factor (LOF) algorithm in the workflow to indicate a local outlier. The descriptions of both the algorithms are provided in the Appendix C and Appendix D. In the comprehensive survey of outlier detection techniques for data streams [28], the authors explain the concept of incremental LOF [29], in which the past data objects together with the local outlier factors are maintained at k. Close to their approach, by utilizing both the variants of the outlier detection techniques, different types of outliers can be obtained. The outlier scores are separately obtained for both the detection algorithms, and there may be no common outlier data objects amongst the two algorithms.

The detection and the voting mechanism involves to detect outliers at each k and subspace by both LOF and RBDA by maintaining a global outlier data store that keeps the count of number of times the data object have the tendency of outlierness, and the outlier scores obtained at each k and subspace. By the intersection of both the data stores, finally the common outliers are produced.

## 6. RQ3: How to Show Similarities and Differences in the EMA of Users Who Are Similar in Their Clinical Data?

For the visualization of similarities and differences, we propose a system as shown in the activity diagram Figure 2, which combines the mechanisms for predictions and for outlierness computation. The figure shows the interactions between the user and the domain expert (physician or clinician) with the system.

### 6.1. Similarity Overview User Panel

An overview of the proposed visual interface is shown in the Figure 3. The design of the visual interface is based upon the visualization guidelines introduced in [30], which is to firstly provide an overview, allowing to zoom to required regions in the visualization, filter, and allowing to provide details on demand. The panel is designed based on iterative discussions and some inspirations from the existing visualization literature. The designed interface is divided into six parts: (1) denotes the selection panel of the selected study user with 1(a) representing the user-identifier; 1(b) denotes the selection of the criteria of similarity; 1(c) is a selection drop-down to choose the question combinations; (2) represents the study user time-series observations displayed over a line chart with capabilities of interactions; (3) shows the similar users to the study user highlighted in orange; (4) represents the comparison of the time series recordings grouped by sessions of the day for the selected user and the nearest neighbor; 4(a) denotes the selection of various attributes to dynamically render the comparison plot; (5) represents the comparisons of the static questions between the study user and the selected nearest neighbor; and (6) represents a slider for allowing to interactively change the number of the nearest neighboring users and updating the visualizations accordingly. Furthermore, we also depict the user panel for EMA (Refer Appendix E
Figure A1).

### 6.2. Nearest Neighbor Visualization

The visualization consists of two steps. At first, through the analysis steps, the required similar user data and information are obtained. Secondly, by using the obtained similar users and the information, color encoding rules are created as represented in Figure 4b and visualized as a graph.

Construction of the visualization: To visualize the nearest neighbors to the study user and to represent the user characteristics, a node-link diagram is utilized as per [31], and the users are placed in the best fit by using the force-directed scheme. However, we add another aspect, namely the representation of each user node with multiple visualization elements.

The Figure 4 shows the nearest neighbor visualization: (1) the study user (orange) and nearest neighbors (blue), (2) the closest neighbor with the edge in bold. The tailored visualization of a single user node is depicted in Figure 5, with the following interactive visualization elements:

Outer Circle: when clicked, the study user is highlighted in orange and circles depicted in blue show the nearest neighbors to the study user.

Inner Circle: the color of the area inside the inner circle is determined by the rules derived from the answers of the user, scaled from 0 to 100 percent. Based on the values, blue color bands ranging from light to dark are used to indicate high to low scale values, represented as shown in Figure 4b. The circle radius is drawn dynamically based on the numerical valued answer of the users. Firstly, a default value for the radius is defined as *r* and based on the answer value, the radius is dynamically updated by adding the answer values to the radius denoted by r:r+answer_val.

Outlier Score Indicator: As outlier scores, we use the z-scores provided by the Rank Based Detection Algorithm (RBDA). To visualize the scores within the nearest neighbor visualization, we introduce a rule-based color coding with values ranging from blue to red, with the shades of red indicating the suspected outliers. The color encoding legend is shown in Figure 4c, representing the color-coding methodology.

Circle Border: the boundary of the circle is colored with purple when the users answered YES to the question “TSCHQ26-Do you think you have a hearing problem?”.

#### Interactions


*Zoom* into each of the user’s nodes to understand the level of the loudness (TSCHQ12) shown in the inner circle and the presence or absence of the hearing problem (TSCHQ26) answer shown in the circle boundary.*Hover* in the center of user node to obtain the user clinical data. (Refer: Figure A2 in Appendix E for more information). By such information easily, a domain expert can identify the differences amongst the obtained similar users.*Click* on one of the nearest neighbors to obtain comparison plots to compare the time series observations as sessions, and the registration properties with the study user.Additionally, the distance information is provided within the nearest neighbor visualization on click, which generates a table (Refer Use Case Example: Figure A4) calculated as the difference in distance of each of the nearest neighbors from the first nearest neighbor. A cell within the generated table is colored pink when the obtained difference in distance value of a nearest neighbor is greater than a threshold (τ) defined as the mean (μ)+1 * standard_deviation (σ) of the distance differences amongst all the nearest neighbors. This provides the information of the dissimilar users in the computed neighborhood.A slider is provided to the domain expert to dynamically increase or decrease the number of nearest neighbors for a study user.


### 6.3. Comparison Visualization

When a click interaction is made from the nearest neighbor visualization, two plots are obtained. The Figure 3(4) shows the comparison between the EMA recordings of the study user and the interacted nearest neighbor user. The plot of study user is colored orange, and the nearest neighbor in blue follows the same coloring standard. The users record their observations multiple times in a day, containing the date and timestamp. From the timestamp, the hour recording value can be utilized to group the sequence of observations into Early Morning (4:00–8:00), Morning (08:00–12:00), Afternoon (12:00–16:00), Evening (16:00–20:00), Night (20:00–00:00) and Late Night (00:00–4:00). By utilizing this information, a box plot is created for each of the user recorded observations and interactions through the tool is provided. These plots are useful in comparing two users who are similar in their static properties may also be similar in their EMA recordings at parts of the day. The Figure 3(5) shows an important plot for the comparison of the static properties of the study user and the interacted nearest neighbor. For representation, the user’s static data are ordinal encoded and presented in the form of a heatmap.

### 6.4. Predictive Visualization over the Line Plot of the Study User

Let the study user trajectory for a variable under consideration be represented by the set of observations $\left[o_{1},o_{2},\ldots ,o_{i},\ldots ,o_{n}\right]$, where *n* is the number of observations and each of the observation $o_{i}$ contain $\left\{t_{i},v_{i}\right\}$, where $t_{i}$ is the aligned time index called as day_session_index and $v_{i}$ is the value of the EMA variable (s01–s08) under consideration. This trajectory is then visualized over the line plot, with the x-axis denoting the time aligned index and the y-axis showing the variable under consideration. This work concentrates on visualizing one of the attributes of the Ecological Momentary Assessment (EMA), which is tinnitus distress (s03); recordings are shown in the Figure 6. To visualize the predictions for the ahead timepoints, a pathway prediction visualization is introduced.

Construction of the pathways: Firstly, the ahead predictions for the reference timepoint are obtained, and then they are dynamically redrawn over the line plot to show the pathways.

#### Interactions


*Click* over each of the data points on the plot and a prediction is made for the next timepoints over a selected data point, and the pathway is visualized in Figure 6(2).*Hover* functionality; compare the predicted with the actual values.*Brush* (Figure 6(1)) a region of timepoint; perform a prediction in that region.


## 7. Results

### 7.1. RQ1: Experimental Setup

The experimental setup involves creating a split of the users into 80%(train) and 20% (test) to investigate the effectiveness of the proposed method for the prediction of the target variable *tinnitus distress-(s03)* from the Ecological Momentary Assessment (EMA) data. A neighborhood is created first by using the static registration data over the defined subspaces, and secondly, by using the Ecological Momentary Assessment (EMA) variable *Loudness-(s02)* for the neighborhood creation amongst users as per the proposed methodology in Section 3.3.2. After the neighborhood creation, a train and test split through randomization is created. For static registration similarity, the registration data are used for the neighborhood creation, and the obtained nearest neighbor time series from the EMA data is used for the ahead predictions of the test users. In the case of loudness-based similarity, both the neighborhood creation and the prediction are over EMA data.

The user’s multiple recordings within the day are aggregated as mean (mean_day), maximum (max_day), and minimum (min_day) observations for the day. This work addresses such a scenario to investigate which of the aggregation function leads to the best prediction of *tinnitus distress-(s03)*.

### 7.2. RQ1: Comparison across Combinations

At first, all the questions were considered for the evaluation. The distance weighting was not useful in the ahead prediction, and hence, was not considered further in our analysis. By utilizing the combination C1–C3, and constructing a neighborhood, it was observed that the combination C3, mean_day aggregation is best in the ahead prediction of tinnitus distress-s03 as shown in the Figure 7. Additionally, for larger neighborhoods, there is not much difference in the error rate of the subspace combinations.

Table 2 provides the average for *k* ranging from 3<=k<=30 for each mean_day, min_day, and max_day observations of the neighbors assisting in the predictions. The lowest average RMSE value obtained across the neighborhood range is highlighted in bold, along with the standard deviation. Combination C3 has the lower RMSE with C3 performing better for both mean_day and min_day with mean_day having low RMSE values.

### 7.3. RQ1: AvgRMSE for Near and Far Predictions

The previous analysis showed that mean_day and combination C3 performs best with respect to the ahead prediction of tinnitus distress. To assess if the ahead predictions remain the same even for the further days for the test users, execution is performed following the same evaluation process by choosing the starting day at far (80%) of the test user’s time series recordings, over the k values, and the subspace combination as C3. A comparison is made over the early and far predictions. The Figure 8 denotes the near and far comparisons of both mean_day aggregations of nearest neighbor time series recordings. It is observed that predictions at earlier days are better than at far days. A possible explanation is that for the far off days there are data from fewer users, thus more noise.

### 7.4. RQ1: Comparison over the Similarity Constructed through Loudness-s02

To compare if the loudness values recorded by the users can achieve better similarity, and in-turn improves the predictions of tinnitus distress-(s03), a neighborhood is constructed, and ahead predictions are evaluated for the earlier days as seen from the Section 7.3. The mean_day aggregations of the nearest neighbors are better (Refer: Appendix A). K values between 5 and 11 have lower RMSE and the error rate increases gradually at the higher neighborhood sizes.

### 7.5. Important Findings of RQ1

As seen from previous results that the RMSE decreases as k increases but at the higher *k* values, there is not much change to the RMSE. So, the best value for the k chosen is at the point where the error rate stops changing or increases (elbow method). Based upon this, it can be seen that *k* = 9 is suitable by considering static registration-based similarity over the combination C3, and mean_day. By considering the loudness-(s02) based similarity, *k* = 11 is suitable with mean_day aggregations of nearest neighbors. Next, two results, one considering the neighborhood over static similarity and the other considering loudness-based similarity, are shown with the obtained values and following the evaluation procedure as per Section 4.2.

By fixing the value of *k* to 9, for the combination-C3 over the static registration similarity, for the loudness-based similarity *k* to 11, and choosing the mean_day observations of the nearest neighbors, 3 ahead predictions are reported by selecting multiple bounds (20%, 30%, 50%) as the start day of the test user’s time series recordings. The RMSE obtained for the ahead predictions at each of the bounds is averaged and results as an average RMSE over multiple bounds. A discretization process is applied for the target variable tinnitus distress-(s03) to assess if the test users in the neighbor ranking positions move similarly to know if the test users have low tinnitus distress values, or high tinnitus distress. The discretization process involves converting the numerical tinnitus distress values into categories. Here, the categories created are [0–0.4] as Low, [0.4–0.7] as Moderate, and [0.7–1.0] as High as per [25]. Some important findings of this analysis are that the test users with the lower RMSE values tend to have most of the observations recorded in the Low category, and the RMSE is higher when the observations are in the High categories. This provides an understanding that the users who have their tinnitus distress in the range of Low to Moderate categories are similar together as others. However, it was also observed that there were few exceptions for certain users whose most observations are recorded in Moderate and High categories. This was further verified by combining (train+test) recordings and was seen that the probability of the observations belonging to the Low category is highest with a value of 0.497 providing an insight that better predictions are obtained when observations fall into the Low category. We tabulate the frequencies and prior probabilities of tinnitus distress against each of the categories in the Table 3.

### 7.6. RQ2: Results of Outlier Detection Mechanisms

#### 7.6.1. Results over the Created Synthetic Data

In this work, the synthesized data are created using PyOD [32] library, where the outlier instances are created by the uniform distribution and normal instances from the Gaussian distribution. The created data is evaluated with Area under the ROC-curve (AUC) [33]. Two scenarios are highlighted over the synthesized data to assess the ability of the Ranked Based Detection Algorithm (RBDA) in comparison with LOF (Local Outlier Factor) to understand the differences in the detection rate as the neighborhood (k) increases.

**Scenario-1:** In this scenario, 100 instances are chosen as a global detection with number of features as 35, k is varied from 10 to 60, and 10 outliers are ingested. The chosen distance measure is Euclidean for both RBDA and LOF. From the Figure 9(Left), both algorithms are able to detect outliers, while LOF detects them at earlier neighborhood size and do not change with increasing the neighborhood size. The RBDA detects outliers at full extent at the higher neighborhood values.

**Scenario-2**: In this scenario, the number of instances are chosen as 100, containing 4 clusters of data points containing 15 features, the cluster have different size and density. The k value is varied from 10 to 60, and 10 outliers are ingested. The chosen distance measure is Euclidean for both RBDA and LOF. From the Figure 9(right), at initial neighborhood sizes, LOF is able to detect the outliers well. However, for neighborhood with larger size k, the detection rate decreases and ranking based detection performs better. This is because of the clusters are of varying densities and the LOF is not be able to identify them for large neighborhood sizes.

From this analysis, we make sure that both of these detection mechanisms can be included in our proposed workflow.

#### 7.6.2. By Using the Tinnitus Data

From the proposed voting-based methodology as per Section 5, differently behaving users are identified from both the introduced similarity methods. The outlier scores from the Ranked Based Outlier Detection obtained from the workflow are used to indicate differently behaving users in the visualization tool (Refer: Section 7.7). By using the same discretization process as per Section 7.5, most of the user’s belonged to either high loudness+high tinnitus distress or low loudness+low tinnitus distress groups. However, there are few users who had high loudness+low tinnitus distress. A plot showing the total number of observations of loudness and tinnitus distress of the users are shown in Appendix A.

### 7.7. RQ3: Validation of the Visual Interface through Usage Scenarios

To validate the effectiveness of the introduced visualizations, various usage scenarios are introduced. For some use cases, simulated data are utilized.

#### 7.7.1. Simulated Users Data Creation for Validation

The simulated users are created by utilizing the same data from the pilot study. We basically create three different type of users: (1) Normal users: these users are created by obtaining a random value for a specified feature from the existing static data and for a given set of a range (in days) generating a random set of timestamped values for creating the EMA recordings; (2) outlying user: the top-n outlying users detected as differently behaving in the interaction of the app from the introduced outlier detection methodology for both static and EMA recordings are used for the study; (3) twin user: the twin user is just the exact copy of one of the selected user from the existing static data and having a randomly generated EMA recordings for a given range.

#### 7.7.2. Outlier User Scenario

Study of a user whose static properties deviate from the rest of the users, also termed as an outlying user. For this scenario, user-20 is studied from the visualization tool. From the tool, combination C4 is chosen for the analysis so that all features can be observed. The same procedure is followed for other subspace combinations as well. The Figure 10 shows the 5 nearest neighbors through the visualization. The outlier score is indicated as a coloring encoding box. More details of the visual explanations are provided in Figure 4. Here, the user-20 is an outlier identified by the drawn dark red rectangle, indicating the higher z-score values obtained from the voting-based method (Refer Appendix E
Figure A5 for the zoomed view).

#### 7.7.3. Twin User Scenario

Study of a user whose static properties are exactly the same as the one of the other user in the underlying dataset. The main focus of this scenario is the understanding and assessment of the constructed similarity in identifying the closest neighbor as the user with exactly the same static properties. From the Figure 11, for the simulated user-444154, the closest neighbor is 154. The simulated user is the *copy of user-154* shown by the last 3 digits of the user-id of the simulated user. This indicates that the proposed user similarity in this work is able to identify it correctly. The distance value will be 0 in this case and from the visualization perspective, an offset of distance is added between the study user and closest neighbor to avoid overlap as shown by the dotted marker line in Figure 11. The distance information is also provided (Refer Appendix E Figure A4).

## 8. Discussion

The previous Section 7 highlighted some important results of the research questions concerning the ability to predict user’s EMA based on similarity, identification of individuals whose properties are different, and the overview of key usage scenarios addressed in the visualization tool.

Regarding the RQ1, from the Figure 7 it confirms that the subspaces do help in the ahead predictions of the user’s EMA recordings as opposed to considering all the questions. Similar behavior was also observed in [34] where the predictive power of the model was assessed through clustering. However, considering only the user’s EMA recordings to construct similarity has a lower error rate as compared to considering the registration data. Therefore, it can be inferred that a similarity measure considering both forms of data together must be explored. As seen from Figure 8, the prediction of user’s EMA for the far future is indeed difficult, which is in agreement with the findings of [25].

Regarding the RQ2, we showed that there exist users whose EMA observations deviate from most of the other users. For example, it is already known that there are users with high tinnitus loudness and low distress [35]. These users behave differently in their interaction with the app, and closer inspection are of interest to the domain expert through interacting with the tool. The predictive mechanism introduced in [34] can be enhanced by including our outlier detection methodology within the workflow.

The proposed medical analytics interactive tool introduces two innovative visualizations regarding RQ3. As seen from the user-scenarios in the Section 7.7, we were able to correctly detect the nearest neighboring user’s based on their daily life behaviors recorded through the app. As seen from the outlier scenario, the visual encoding of the color code provides useful information to the physician about a user’s outlierness. The proposed ahead prediction visualization can help in reconsidering the patient treatment or monitoring policy.

Finally, the results are briefly reflected in a broader scope. In general, we have learned in recent years that mHealth can deliver interesting results for medical purposes [36]. However, the use of mHealth has revealed too many flavors without comparable construction principles like presented by [37]. Consequently, we have to investigate comparable and powerful tools to approximate ourselves to tangible medical results based on the use of mHealth. The idea to compare users might open one more perspective to learn more about mHealth behaviour and users. In this light, with this work, we may enable both computer scientists to create new algorithms better and empower medical experts to more quickly reveal user differences. As tinnitus is a very heterogeneous phenomenon, works like this may help to better demystify this heterogeneity.

From a practitioner point of view, the proposed method could be useful to identify those patients that need special care or will need additional special care any time soon. Especially in the case of sparse resources, it would allow focusing on those patients that need more or specific care to prevent more severe clinical conditions (and associated costs). By preventing higher costs and severe conditions, a commercial solution could gain market share over others that were not able to predict future data based on currently available patient information.

**Threats to validity:** The experimental design for RQ2 did not contain class labels to verify whether a user is behaving differently or not. To truly assess RQ2, an experiment setup must be made over a labeled dataset and verified. The underlying data did not have user answers for Mini-TQ questionnaire, and we firmly believe that making this mandatory in the app can help to better assess the user’s EMA recordings [34]. In the proposed interactive tool, our ahead prediction visualization is concerned with only one of the EMA variables. Including predictions for the other EMA variables will be of greater benefit to the physician. The use cases explored in our interactive tool were developed with limited input from very few physicians, and a wider user study with more experts would give better insights into whether the use cases are exhaustive and sufficient from the point of view of a physician.

**Future actions:** As part of the future work, we are looking forward to exploring and exploiting similarity methods that can work on both static and dynamic data, and perform a comparison experiment on our introduced and existing approaches. In terms of the visualization, we look to explore new approaches of visualization to compare EMA recordings as event sequences.

## 9. Conclusions

To conclude, we showed from the tool that it is possible to predict a user’s future EMA based on the data of similar users, with the ability to identify outlying individuals. Our findings on outlierness indicate how important it is to closely monitor the users’ EMAs as a medical practitioner, who can assess whether action is needed as EMA values change. Our findings on the evolution of users with similar registration data over time show that registration data alone are not adequate to assess similarity during EMA recordings. Apart from the quality of predictions, we also see that the results of such an interactive system to explore the usefulness of neighborhoods and the outlierness of an individual can help bring scientists in two fields come closer together. The medical practitioner can use such a system to better intuit factors that make users different, which researchers in computer science can use to develop better algorithms to discover these differences and visualize them in a way that non-experts can intuitively understand.

On a technical front, we discuss some opportunities for extensions on both the prediction of EMAs in the near and far future, and also the possibility to simultaneously exploit static and dynamic data during the neighborhood discovery process. The results of these methods can be compared against the current proposed method, which can serve as a baseline. The design of the visualization can be adapted to this special case with minimal effort. We also make an initial attempt with the facilitation for collecting tinnitus static data (Refer: Figure A3). We further look to improvise regarding automatic collection, information processing and visualize within the tool itself.

More work is needed to understand whether there are subpopulations of users who have similar registration data and evolve similarly. Our findings on near and far future prediction of EMA indicate that after the first days of interaction with the mHealth app, the involvement of the users may decrease. Hence, mHealth app developers and medical practitioners should think of incentives to stimulate regular interaction (for example, gamification).

## Figures and Tables

**Figure 1 entropy-23-01695-f001:**
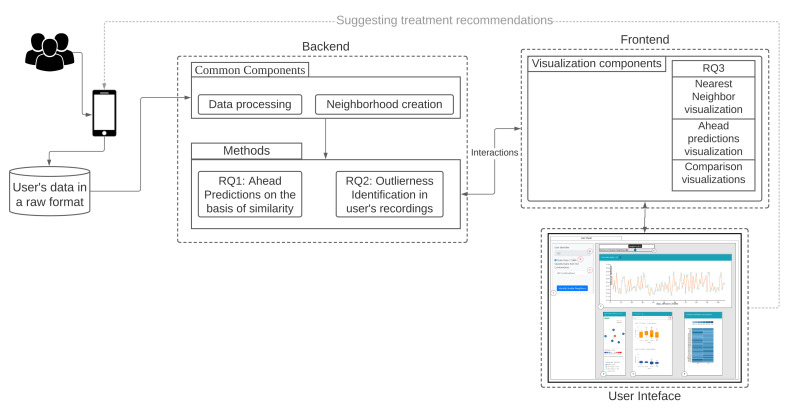
Overview of methods and components of our approach.

**Figure 2 entropy-23-01695-f002:**
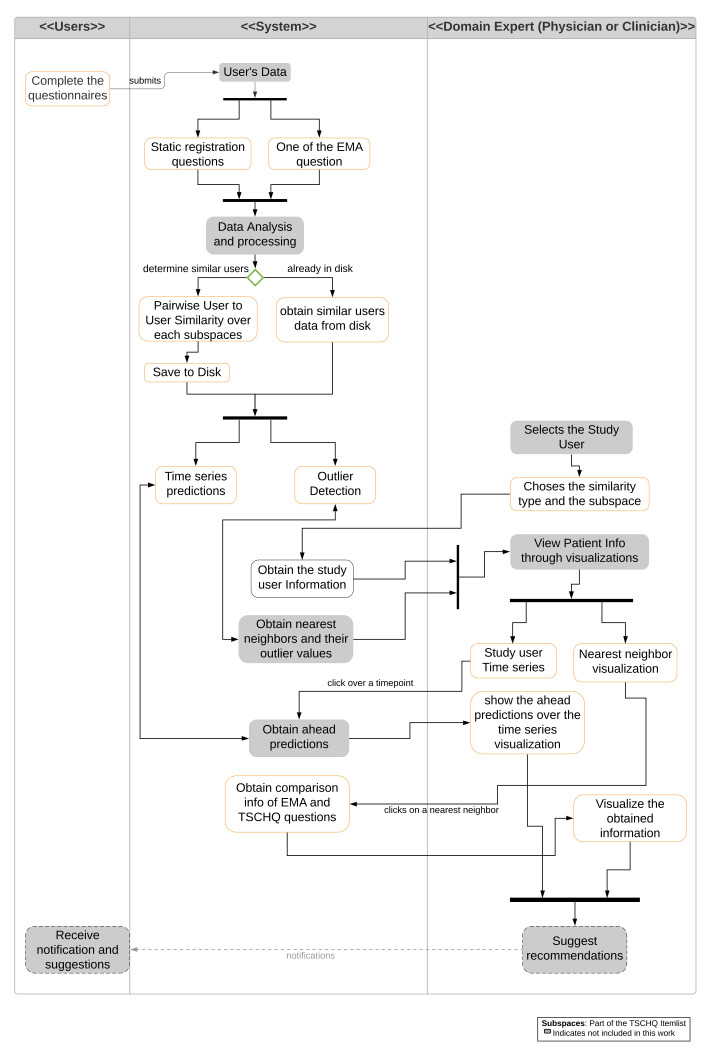
Activity Diagram of proposed visualization system. Main interactions with users (**left**), system (**center**), and expert (**right**) are represented. Various functionalities and actions are shown. Gray boxes indicate actions not included as part of this work.

**Figure 3 entropy-23-01695-f003:**
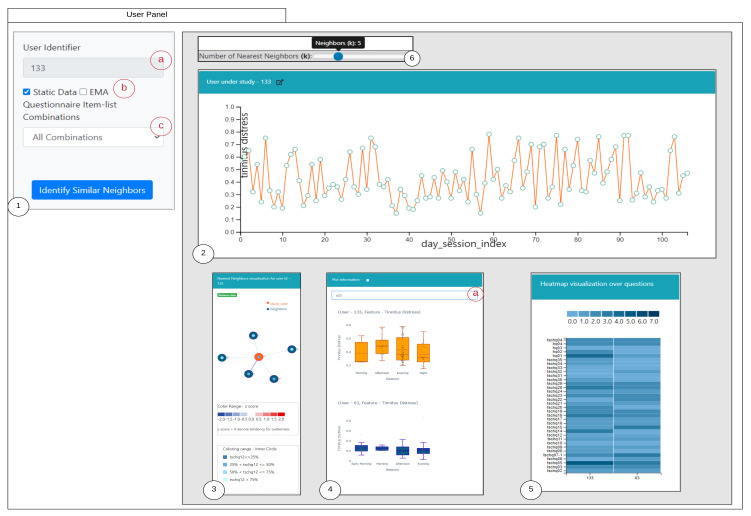
Similarity Overview User Panel.

**Figure 4 entropy-23-01695-f004:**
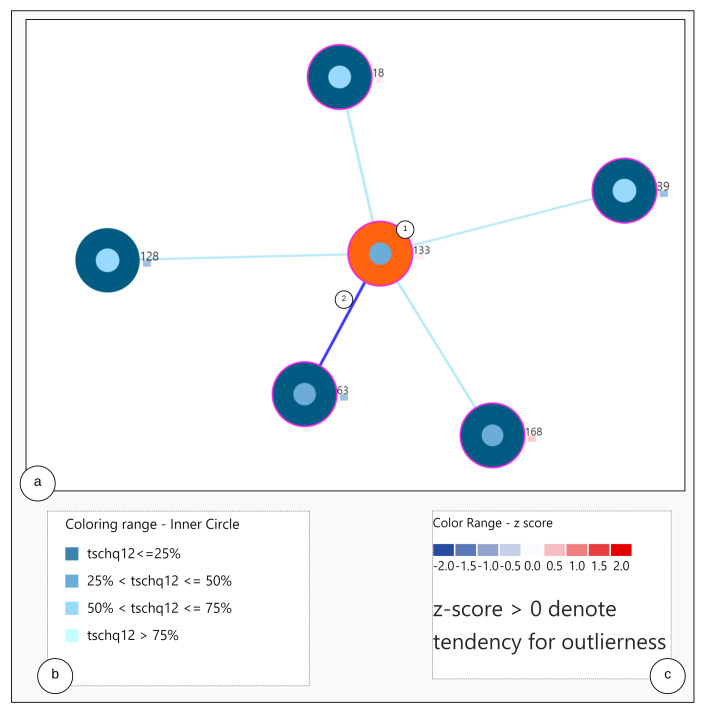
Nearest Neighbor Visualization.

**Figure 5 entropy-23-01695-f005:**
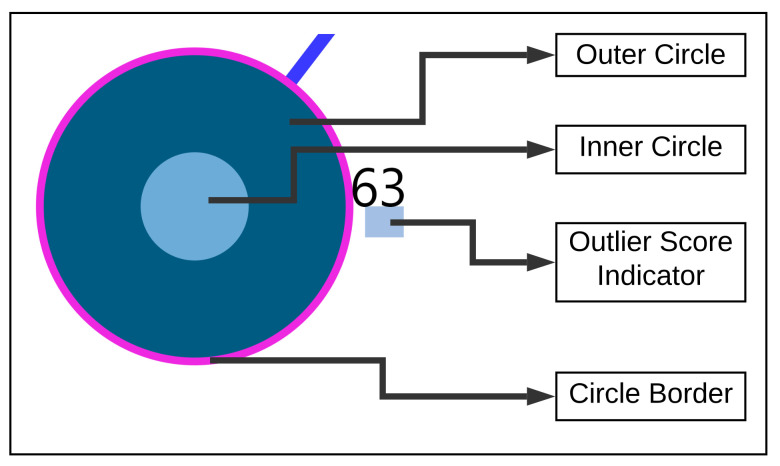
Tailored visualization for each user.

**Figure 6 entropy-23-01695-f006:**
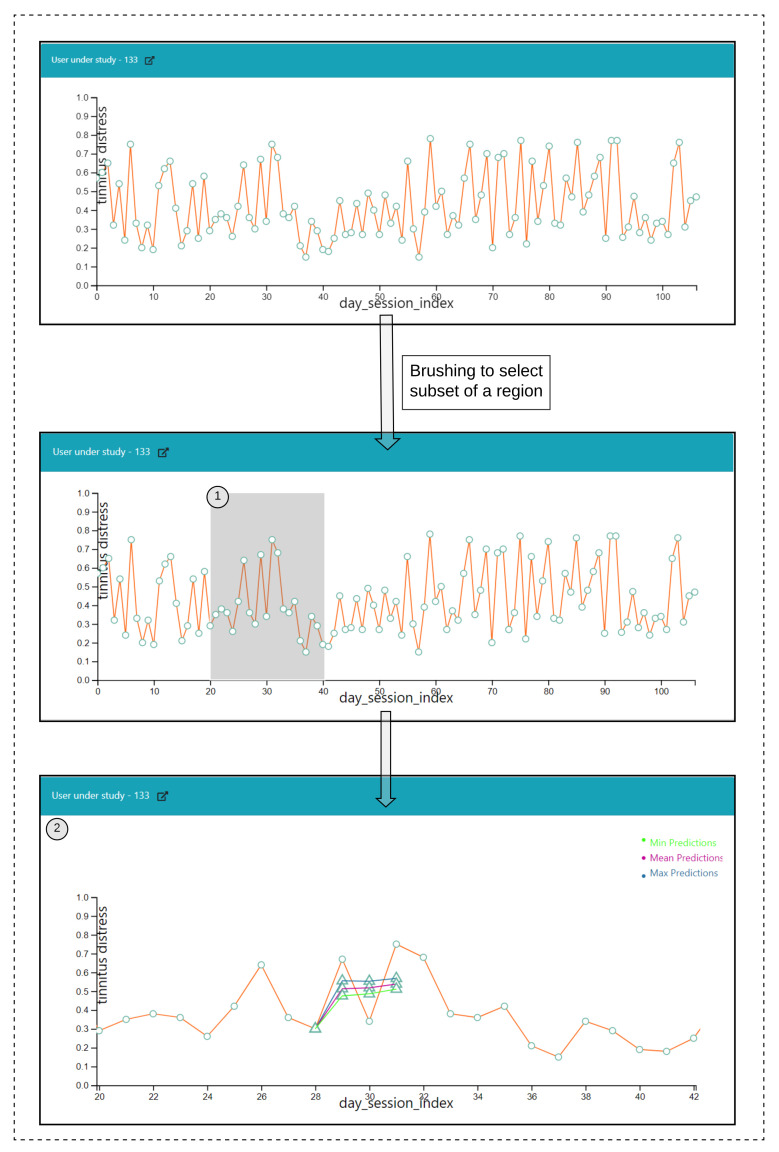
Visualization to depict predictions.

**Figure 7 entropy-23-01695-f007:**
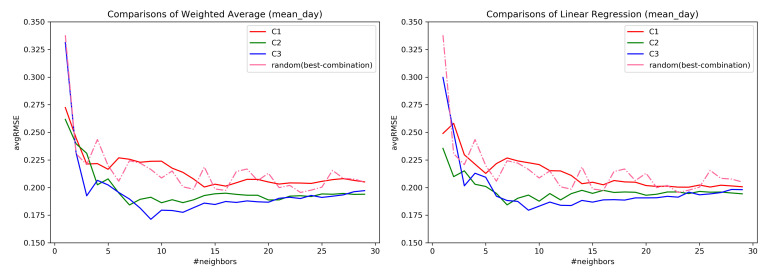
Prediction comparison over mean_day observations of users.

**Figure 8 entropy-23-01695-f008:**
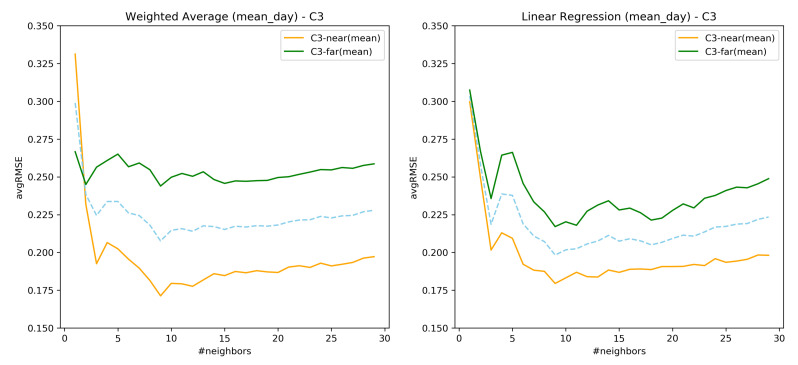
Near and Far future predictions.

**Figure 9 entropy-23-01695-f009:**
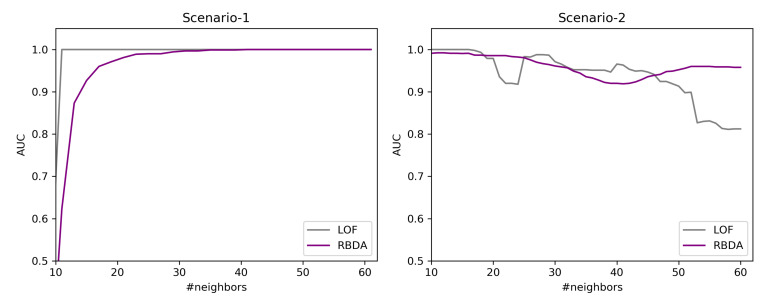
Simulated data: different scenarios for outlier detection.

**Figure 10 entropy-23-01695-f010:**
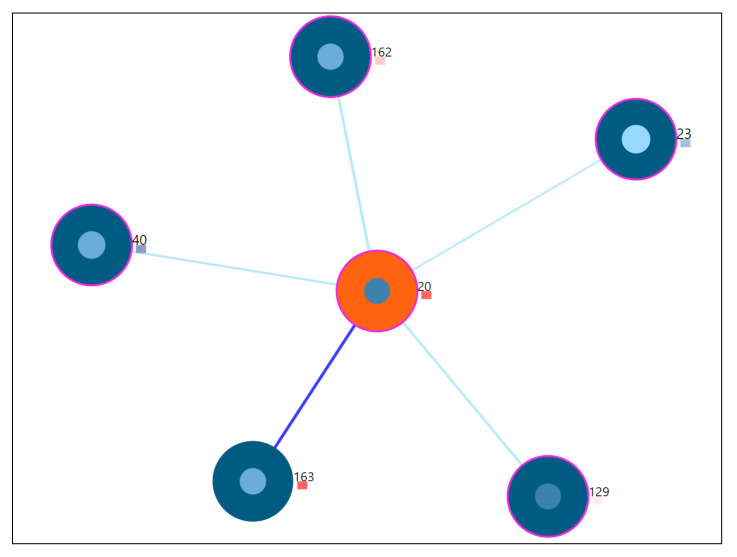
5 nearest neighbors for outlier user-20.

**Figure 11 entropy-23-01695-f011:**
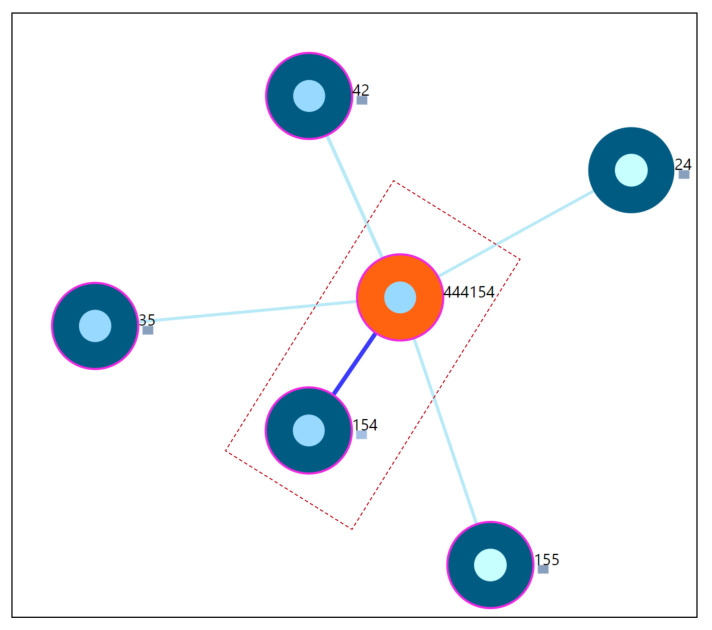
Nearest neighbors of simulated user-444154.

**Table 1 entropy-23-01695-t001:** Overview of notation used in following sections.

Notation	Description
*u*	a user of the application
*x* and *z*	Whenever a sentence refers to more than one user, these users are denoted them as denote as x,z
*U*	the set of all users who use the mHealth application, which is the data source of this study
*D*	the (sub)set of users used for training the models; D⊆U
*T*	the (sub)set of test users, who are not used during model training, but are used to assess model quality; T⊆U
*k*	neighborhood size
$NN_{k}\left(u\right)$	the *k* nearest neighbors of *u*
staticF	the feature space of the registration data, with $s_{r}$ being a feature, for r=1,…,|staticF|
*F*	the feature space of EMA recordings, with $f_{j}$ being a feature, for j=1,…|F|; |F|=8
$t_{i}$	a timepoint
$f_{i}$	A feature belonging to EMA feature space *F*
l≥1	the timepoints in the future for which we make predictions, i.e., we predict $o_{i+l}$ at $t_{i}$ for user *u*
$\hat{f_{i}}$	A predicted value for feature $f_{i}\in F$

**Table 2 entropy-23-01695-t002:** Average of WA—weighted average; LR—linear regression over combinations.

Feature Combinations	Max_DAY		Mean_DAY		Min_DAY	
	WA	LR	WA	LR	WA	LR
C1	0.23407	0.2332	0.21105	0.2096	0.20481	0.20414
	±0.00656	±0.0059	±0.0084	±0.00948	±0.00857	±0.01003
C2	0.2156	0.2188	0.19403	0.19539	0.19252	0.1973
	±0.00808	±0.0048	±0.00859	±0.00529	±0.00899	±0.00454
C3	0.2077	0.2099	**0.18882**	**0.19191**	**0.18948**	**0.19619**
	±0.00925	±0.00923	**±0.00745**	**±0.007320**	**±0.0116**	**±0.0122**

**Table 3 entropy-23-01695-t003:** Frequencies and prior probabilities of target variable tinnitus distress-(s03).

Category	Num Users	Total Number of Observations (s03)	Probability
Low	45	1911	**0.497**
Moderate	48	1155	0.30
High	37	773	0.201
Total	-	3839	1.0

## Data Availability

The data used in this study are not available publicly. The authors may be contacted in case of data requests.

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
