# Peer review of "Interactive System for Similarity-Based Inspection and Assessment of the Well-Being of mHealth Users"

_entropy, 2021, doi:10.3390/e23121695_

Round 1

Reviewer 1 Report

I have read this paper in detail.

Based upon my careful observation, the work and the presented results are good. Therefore, it can be published subject to the following major modifications/corrections/queries:

-The conclusion section should be extended
-The main motivation of the paper should be explained in detail.
-Numerical simulations should be discussed clearly
-The references list should be updated

-Conclusion section is too short. They need to extend this part of the paper by giving novelty of the paper

-Details on the future implementation are needed

-Conclusion section should be rearranged and increased. According to the topic of the paper, the authors may propose some interesting problems as future work in the conclusion.

Explain your original contribution.

1) The organization of this paper is appropriate. However, authors should polish the paper suitably.
2) The introduction is too short and then it should be expended and reconstructed. Including the motivation, the main difficulties, the main work and the improvements compared with previous related works should be emphasized in this section.
3) The novelty of the proposed method should be highlighted carefully.
4) Recent published papers dealing with the Ecological Momentary Assessments must be considered

5) The importance of the problem considered in this paper should be further addressed.
6) You should focus on it to further present your contributions in this work.
7) Authors should give a Remark to illustrate how predict the condition of the user and also identifying the clinical data. 

Reviewer 2 Report

Thank you for submitting your work to the journal. I've read it with interest. Generally, I am impressed by the quality of work and the way it is presented. Overall, I am happy with the paper and can recommend it for publication with some revisions. 

-Literature review needs to be done more thoroughly and comprehensively. I see many important relevant works are not cited nor considered critically.

The introduction section is written clearly but it could have been better with a clear identification of research gaps. It is not sure what the research gaps are and how you fill out the gaps. Based on the gaps, you should clearly state the goal of the study. 

Table 1 is hard to understand.

The method section can be further detailed with clear processes and how you collected the data in the study.

Discussion can be enhanced. In its current form, discussion or in-depth discussion is lacking.

Round 2

Reviewer 1 Report

Accept the paper

Author Response

We were a bit uncertain as to how to address this comment, since it doesnt point us to any section in particular that is difficult to understand.

We have therefore done another reading pass and a  check with grammarly-like software.

This manuscript is a resubmission of an earlier submission. The following is a list of the peer review reports and author responses from that submission.